META-RESEARCH

# Use of the Journal Impact Factor in academic review, promotion, and tenure evaluations

**Abstract** We analyzed how often and in what ways the Journal Impact Factor (JIF) is currently used in review, promotion, and tenure (RPT) documents of a representative sample of universities from the United States and Canada. 40% of research-intensive institutions and 18% of master's institutions mentioned the JIF, or closely related terms. Of the institutions that mentioned the JIF, 87% supported its use in at least one of their RPT documents, 13% expressed caution about its use, and none heavily criticized it or prohibited its use. Furthermore, 63% of institutions that mentioned the JIF associated the metric with quality, 40% with impact, importance, or significance, and 20% with prestige, reputation, or status. We conclude that use of the JIF is encouraged in RPT evaluations, especially at research-intensive universities, and that there is work to be done to avoid the potential misuse of metrics like the JIF.
DOI: https://doi.org/10.7554/eLife.47338.001

**ERIN C MCKIERNAN[†*], LESLEY A SCHIMANSKI, CAROL MUÑOZ NIEVES, LISA MATTHIAS, MEREDITH T NILES AND JUAN P ALPERIN[†*]**

**\*For correspondence:**
emckiernan@ciencias.unam.mx
(EC); juan@alperin.ca (JP) (ECMK);
juan@alperin.ca (JPA)

[†]These authors contributed
equally to this work

**Competing interest:** See
page 10

**Reviewing editor:** Emma
Pewsey, eLife, United Kingdom

## Introduction

The Journal Impact Factor (JIF) was originally developed to help libraries make indexing and purchasing decisions for their journal collections (*Garfield, 2006*; *Archambault and Larivière, 2009*; *Haustein and Larivière, 2015*), and the metric's creator, Eugene Garfield, made it clear that the JIF was not appropriate for evaluating individuals or for assessing the significance of individual articles (*Garfield, 1963*). However, despite this and the various well-documented limitations of the metric (e.g., *Seglen, 1997*; *Moustafa, 2015*; *Brembs et al., 2013*; *The PLOS Medicine Editors, 2006*; *Kurmis, 2003*; *Sugimoto and Larivière, 2018*; *Haustein and Larivière, 2015*; *The Analogue University, 2019*), over the past few decades the JIF has increasingly been used as a proxy measure to rank journals – and, by extension, the articles and authors published in these journals (*Casadevall and Fang, 2014*). The

association between the JIF, journal prestige, and selectivity is strong, and has led academics to covet publications in journals with high JIFs (*Harley et al., 2010*). Publishers, in turn, promote their JIF to attract academic authors (*Hecht et al., 1998*; *Sugimoto and Larivière, 2018*; *SpringerNature, 2018*).

In some academic disciplines, it is considered necessary to have publications in journals with high JIFs to succeed, especially for those on the tenure track (for review see *Schimanski and Alperin, 2018*). Institutions in some countries financially reward their faculty for publishing in journals with high JIFs (*Fuyuno and Cyranoski, 2006*; *Quan et al., 2017*), demonstrating an extreme but important example of how this metric may be distorting academic incentives. Even when the incentives are not so clear-cut, faculty still often report intense pressure to publish in these venues (*Harley et al., 2010*; *Walker et al.,*

2010; Tijdink et al., 2016). Concerns about the JIF and journals' perceived prestige may also limit the adoption of open access publishing (Schroter et al., 2005; Swan and Brown, 2004; University of California Libraries, 2016), indicating how the effects of the JIF permeate to the broader scholarly publishing ecosystem.

This use – and potential misuse – of the JIF to evaluate research and researchers is often raised in broader discussions about the many problems with current academic evaluation systems (Moher et al., 2018). However, while anecdotal information or even formal surveys of faculty are useful for gauging the JIF's effect on the academic system, there is still a lot we do not know about the extent to which this metric is used in formal academic evaluations. To our knowledge, there have been no studies analyzing the content of university review, promotion, and tenure (RPT) guidelines to determine the extent to which the JIF is being used to evaluate faculty, or in what ways. We therefore sought to answer the following questions: (1) How often is the JIF, and closely related terms, mentioned in RPT documents? (2) Are the JIF mentions supportive or cautionary? (3) What do RPT documents assume the JIF measures?

## Methods

### Document collection
This paper reports a set of findings from a larger study (Alperin et al., 2019) for which we collected documents related to the RPT process from a representative sample of universities in the United States and Canada and many of their academic units. A detailed description of the methods for selecting institutions to include in our sample, how we classified them, how we collected documents, and the analysis approach is included in Alperin et al. (2019) and in the methodological note accompanying the public dataset (Alperin et al., 2018). Briefly, we created a stratified random sample based on the 2015 edition of the Carnegie Classification of Institutions of Higher Education

(Carnegie Foundation for the Advancement of Teaching, 2015) and the 2016 edition of the Maclean's University Rankings (Rogers Digital Media, 2016), which respectively group US and Canadian universities into those focused on doctoral programs (i.e., research intensive; R-type), those that predominantly grant master's degrees (M-type), and those that focus on undergraduate programs (i.e., baccalaureate; B-type). We used a taxonomy developed by the National Academies in the United States (The National Academies of Sciences, Engineering and Medicine, 2006) to classify the academic units (e.g., department, school, or faculty) within an institution into three major areas: Life Sciences (LS); Physical Sciences and Mathematics (PSM); and Social Sciences and Humanities (SSH). Additional units that could not be classified as belonging to a single area (e.g., a College of Arts and Sciences) were designated as multidisciplinary. The stratified sample was designed to collect documents from enough institutions in each of the R-, M-, and B-type categories to have a statistical power of .8 (assuming a small effect size of .25 of a standard deviation) when making comparisons between categories. An overview of the population of universities by type, the number and percent randomly chosen for our stratified sample, and the number of institutions for which we obtained at least one relevant document can be found in Table 1. A more detailed table, including institution sub-types, can be found in Alperin et al. (2019).

We then used a combination of web searches, crowdsourcing, and targeted emailing to request documents related to the RPT process, including but not limited to collective agreements, faculty handbooks, guidelines, and forms. Some of these documents applied to the institution as a whole, while others applied only to specific academic units. In the end, we obtained 864 documents related to the RPT process of 129 universities, of which 57 were R-type, 39 were M-type, and 33 were B-type institutions. Of the total documents, 370 were

**Table 1.** Sampling summary of universities from Canada and the United States.

|  | Number in category | Number sampled | Percent sampled | Number with documents |
|---|---|---|---|---|
| R-type | 350 | 65 | 19% | 57 |
| M-type | 847 | 50 | 6% | 39 |
| B-type | 602 | 50 | 8% | 33 |

institutional-level documents, while the remaining 494 came from 381 academic units within 60 of these universities. Of the 116 units at R-type institutions, 33 (28%) were LS units, 21 (18%) were PSM units, 39 (34%) were SSH units, and 23 (20%) were multidisciplinary units.

### Document analysis and coding terminology

The RPT documents were loaded into QSR International's NVivo 12 qualitative data analysis software, where text queries were used to identify documents that mention specific terms. Because the language in RPT documents varies, we first searched all the documents for the words "impact" and "journal", and read each mention to identify terms that may be referencing the JIF. We classified these terms into three groups: (1) direct references to the JIF as a metric; (2) those that reference journal impact in some way; and (3) indirect but possible references to the JIF. In the first group, we included the terms "impact factor", "impact score", "impact metric", and "impact index". In the second group, we included the terms "high-impact journal", "impact of the journal", and "journal('s) impact". The third group contains a larger number and variety of terms, such as "high-ranking journal", "top-tier journal", and "prestigious journal". For all terms, we considered both singular and plural equivalents. A map of the terms we found and their grouping into the three categories can be seen in *Figure 1*. In our analysis, we looked at only the first two groups of terms, as we considered them to be unambiguously about the JIF (group 1) or sufficiently close to the notion of JIF (group 2). The terms in the third group, however, may or may not refer to the JIF. So while these terms could represent examples of ways in which the idea of the JIF is invoked without being explicit, their mentions were not analyzed further for this study.

The results of each text query for the terms in groups 1 and 2 were placed in an NVivo "node" that contained the text surrounding each of the mentions. We then performed a "matrix coding query" to produce a table with institutions and academic units as rows, terms of interests as columns, and a 1 or a 0 indicating whether the institution or academic unit made mention of the term or not, with the ability to distinguish if the mention appeared in documents that pertain to the whole institution, to one or more academic units, or both. We considered an institution as making mention of a term if the term was present in at least one document from that institution or any of its academic units.

More details on this process can be found in *Alperin et al. (2019)*.

### Qualitative analysis

We also exported the content of each node for a qualitative analysis of the JIF mentions. In some cases, the software extracted complete sentences, while in other cases it pulled only fragments and we retrieved the rest of the text manually to provide better context. Based on a detailed reading of the text, we classified each of the JIF mentions along two dimensions. First, we classified each mention as either: (1) *supportive* of the JIF's use in evaluations; (2) *cautious*, meaning the document expresses some reservations about the use of the JIF in evaluations; or (3) *neutral*, meaning the mention was neither supportive nor cautious, or not enough information was present in the document to make a judgement. In addition, we read each mention to determine what aspects of research were being measured with the JIF, if specified. Using categories we arrived at inductively, we classified each mention of the JIF as associating the metric with one or more of the following: (i) quality of the research and/or journal; (ii) impact, importance, or significance of the research or publication; (iii) prestige, reputation, or status of the journal or publication; or (iv) left unspecified, meaning the document mentions the JIF, but does not state what the metric is intended to measure. If an institution contained multiple mentions (for example, in two different academic units), it was counted under all the relevant categories.

To arrive at the classification, each mention was independently coded by two of the authors (EM and LM) using the definitions above. After an initial pass, the two coders agreed on all of the classifications for 86% of all mentions. The remaining mentions were independently coded by a third author (LS). In all instances, the third coder agreed with one of the previous two, and this agreement was taken as the final code.

### Data availability

We have shared the data on which this paper is based in two different formats: (1) a spreadsheet with all the JIF-related mentions (including repetitions) extracted from the RPT documents, available as part of the larger public dataset (*Alperin et al., 2018*), and (2) a text document containing the mentions (minus repetitions), with terms of interest color coded and a qualitative assessment of each quote, available as

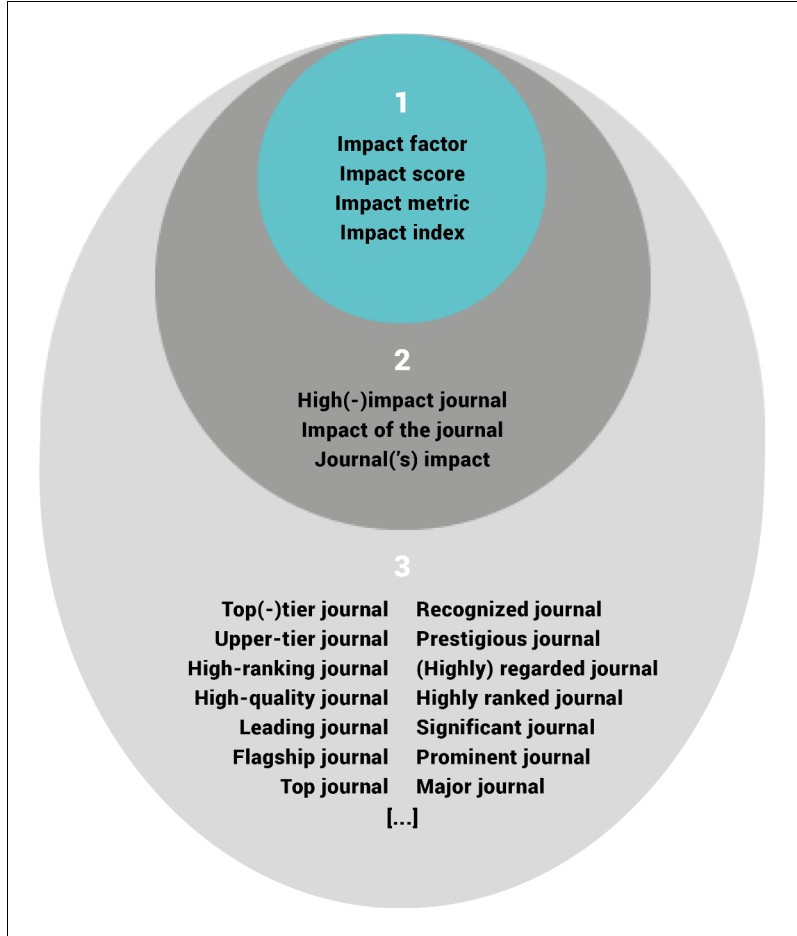

**Figure 1.** Grouping of terms related to the JIF.  Terms found in RPT documents were classified as either: (1) referring directly to the JIF (inner ring); (2) referring in some way to journal impact (middle ring); or (3) indirect but probable references to the JIF. For simplicity, singular versions of each term are shown, but searches included their plural equivalents. Our analysis is based only on those terms found in groups 1 and 2 (the two innermost rings).
DOI: https://doi.org/10.7554/eLife.47338.002

created them. However, for publicly available documents, we included Wayback Machine web archive links to them in the shared spreadsheet.

*Limitations*

Our study covers a broad range of document types that spans an equally diverse range of institutions and academic units. Although we believe the documents analyzed are representative of what is used in practice in RPT evaluations, the diversity of forms and practices means that some documents contain more details than others regarding what is expected of faculty. As a result, the lack of presence of the JIF-related terms may be due to the types of document used at those institutions, and not a lack of interest or focus on using the metric for evaluation.

Along the same lines, we must also recognize that in studying the RPT process through a document-centric approach, our analysis remains limited to what is formalized in the documents themselves. It cannot tell us how RPT committees use the JIF or other citation metrics, if at all, during the process, nor how faculty use these guidelines in preparing their dossiers for review. To this end, we echo the call of *O'Meara (2002)* and our own previous study (*Alperin et al., 2019*) for more work that studies the relationship between RPT guidelines and faculty behaviors, while offering this empirical analysis of RPT documents as foundational evidence.

## Results

### How often is the JIF mentioned in RPT documents?

While metrics in general are mentioned in RPT documents from 50% of institutions in our sample (*Alperin et al., 2019*), only 23% mentioned the JIF explicitly or used one of the JIF-related terms (see groups 1 and 2 in *Figure 1*) in their RPT documents. The percentage was higher for R-type institutions (40%) than for either M-type (18%) or B-type (0%) institutions (*Table 2*). Some mentions were found in the institutional-level documents, while others were found at the level of the academic unit (e.g., college, school, or department). Many of the mentions were from different academic units within the same university. Within the R-type institutions, the percentage of academic units that mentioned JIF-related terms was higher for LS (33%) and PSM (29%) units than for SSH (21%) or multidisciplinary units (17%).

supplemental information. The main data file in *Alperin et al. (2018)* (scholcommlab-rpt-master-april-2019.tab) contains two columns for the JIF (*metrics_impact_factor* and *metrics_high_impact_journals*). A 1 in these columns indicates that at least one document from that institution or any of its academic units contained a JIF term from groups 1 or 2 (*Figure 1*), respectively, while a 0 indicates no such terms were found in any of the documents for that institution. A set of columns with the prefix *if_* similarly contain a 1 if the JIF mention was coded for each category, and a 0 otherwise. We are not able to share the original RPT documents collected for this study, since the copyrights are held by the universities and academic units that

**Table 2.** Mentions of the JIF in RPT documents, overall and by institution type.

| | | All | R-type | M-type | B-type |
|---|---|---|---|---|---|
| How many institutions mention the JIF? | n | 129 | 57 | 39 | 33 |
| | JIF mentioned | 30 (23%) | 23 (40%) | 7 (18%) | 0 (0%) |
| Are the JIF mentions supportive or cautionary? | n | 30 | 23 | 7 | 0 |
| | supportive | 26 (87%) | 19 (83%) | 7 (100%) | - |
| | cautious | 4 (13%) | 3 (13%) | 1 (14%) | - |
| | neutral | 5 (17%) | 4 (17%) | 1 (14%) | - |
| What do institutions measure with the JIF? | n | 30 | 23 | 7 | 0 |
| | quality | 19 (63%) | 14 (61%) | 5 (71%) | - |
| | impact/importance/significance | 12 (40%) | 8 (35%) | 4 (57%) | - |
| | prestige/reputation/status | 6 (20%) | 5 (22%) | 1 (14%) | - |
| | unspecified | 23 (77%) | 17 (74%) | 6 (86%) | - |

Note that percentages do not sum to one hundred in any given column, since many institutions had more than one JIF mention that could be classified differently. For example, an institution was marked as having a supportive mention if at least one RPT document from that institution, or any of its academic units, had a supportive mention. The same institution could also be counted under 'cautious' if a different academic unit within that institution had such a mention.

### Are the JIF mentions supportive or cautionary?

The majority of mentions of the JIF were supportive of the metric's use in evaluations. Overall, 87% of institutions that mentioned the JIF did so supportively in at least one of their RPT documents from our sample (*Table 2*). Breaking down by institution type, 83% of R-type and 100% of M-type institutions had supportive mentions. In contrast, just 13% of institutions overall had at least one mention which expressed caution about using the JIF in evaluations. Two institutions (University of Central Florida and University of Guelph) had both supportive and cautious mentions of the JIF, but originating from different academic units. Overall, 17% of institutions had at least one neutral mention. Examples of supportive and cautious mentions can be found in the following two sections. Examples of neutral mentions are in the supplemental information.

### What do RPT documents assume the JIF measures?
Associating the JIF with quality
The most commonly specified association we observed in these RPT documents was between the JIF and quality, seen in 63% of institutions overall (*Table 2*). By institution type, 61% of R-type and 71% of M-type institutions in our sample that mention the JIF associate the metric with quality. This association can be seen in the guidelines from the Faculty of Science at the

University of Alberta that state: "Of all the criteria listed, the one used most extensively, and generally the most reliable, is the quality and quantity of published work in refereed venues of international stature. Impact factors and/or acceptance rates of refereed venues are useful measures of venue quality..." (*University of Alberta, 2012*).

While some RPT documents recommend using the JIF to determine the quality of a journal, others suggest that this metric can be used to indicate the quality of individual publications. An example of the latter comes from the Department of Political Science, International Development, and International Affairs at the University of Southern Mississippi: "Consideration will be given to publication quality as measured by the following items (though not exclusive of other quality measures not listed here): journal/press rankings, journal/press reputation in the field, journal impact factors, journal acceptance rates, awards, citations, reviews and/or reprints" (*University of Southern Mississippi, 2016*).

Other guidelines create their own metrics using the JIF in their calculations and suggest this will incentivize high quality research, as seen in the following example from the Institute of Environmental Sustainability at Loyola University: "For promotion to Professor, the candidate must have an average publication rate of at least one article per year published in peer-reviewed journals in the five-year period preceding the

application for promotion. These articles should be regularly cited by other researchers in the field. We will consider both the quality of the journal (as measured by the journal's impact factor, or JIF) as well as the number of citations of each publication. We will employ the metric: Article Impact Factor (AIF) = (JIF * citations) where "citations" represents the number of citations for the particular publication. Employing this metric, faculty have incentive to publish in the highest quality journals (which will increase the JIF) and simultaneously produce the highest quality research manuscripts, potentially increasing the number of citations, and increasing the AIF" (*Loyola University Chicago, 2015*).

In sum, there are repeated links made in the sampled RPT documents between the JIF and research, publication, or journal quality.

## Associating the JIF with impact, importance, or significance

The second most common specified association we observed in these RPT documents was between the JIF and the impact, importance, or significance of faculty research or publications, found in 40% of institutions in our sample. By institution type, 35% of R-type and 57% of M-type institutions made this association (*Table 2*). For example, guidelines from the Department of Psychology at Simon Fraser University link the JIF with impact: "The TPC [Tenure and Promotion Committee] may additionally consider metrics such as citation figures, impact factors, or other such measures of the reach and impact of the candidate's scholarship" (*Simon Fraser University, 2015*).

Promotion and tenure criteria from the University of Windsor link the JIF to publication importance (*University of Windsor, 2016*): "Candidates will be encouraged to submit a statement that explains the importance of their publications, which may include factors such as journal impact factors, citation rates, publication in journals with low acceptance rates, high levels of readership, demonstrated importance to their field."

Guidelines from the Department of History at the University of California, Los Angeles associate the JIF with significance of faculty work: "The [policy on academic personnel]'s concern that the candidate be "continuously and effectively engaged in creative activity of high quality and significance," should further be demonstrated through other publications that include peer reviewed articles in high impact journals. . .".

In all of the above cases, the value of faculty research or individual publications is being evaluated, at least in part, based on the JIF.

## Associating the JIF with prestige, reputation, or status

A third set of mentions of the JIF associated the metric with prestige, reputation, or status, typically referring to the publication venue. Overall, 20% of institutions in our sample that mentioned the JIF made such an association. As with other concepts, there was variability by institution type, with 22% of the R-type and 14% of the M-type having at least one instance of this association (*Table 2*). For example, guidelines from the Department of Sociology at the University of Central Florida link the JIF with prestige: "It is also true that some refereed journal outlets count for more than others. Publication in respected, highly cited journals, that is, counts for more than publication in unranked journals. The top journals in sociology and all other social sciences are ranked in the Thompson/ISI citation data base (which generates the well-known Impact Factors), in the Scopus data base, and in certain other citation data bases. In general, it behooves faculty to be aware of the prestige rankings of the field's journals and to publish in the highest-ranked journals possible. It is also advisable to include in one's tenure and promotion file information about the Impact Factors or related metrics for the journals where one's papers appear" (*University of Central Florida, 2015*).

Similarly, promotion and tenure forms from the University of Vermont associate the JIF with journal status: "List all works reviewed prior to publication by peers/editorial boards in the field, such as journal articles in refereed journals, juried presentations, books, etc. Indicate up to five of the most important contributions with a double asterisk and briefly explain why these choices have been made. Include a description of the stature of journals and other scholarly venues and how this is known (e.g., impact factors, percentage of submitted work that is accepted, together with an explanation of the interpretation of these measures)" (*University of Vermont, 2016*).

Overall, these documents show a focus on publication venue and use the JIF as a proxy measure for determining how much individual publications should count in evaluations based on where they are published.

## Many mentions do not specify what is measured with the JIF

Lastly, we were left with many instances where the JIF was mentioned without additional information on what it is intended to measure. Such unspecified mentions were found in the RPT documents of 77% of institutions that mentioned the JIF. These correspond to 74% of the R-type institutions and 86% of the M-type institutions with mentions (*Table 2*). These mentions were often found in research and scholarship sections that ask faculty to list their publications and accompanying information about the publication venues, such as the JIF or journal rank. Some of these documents simply suggest the JIF be included, while others make it a requirement. For example, guidelines from the Russ College of Engineering and Technology at Ohio University request the JIF in the following way: "List relevant peer-reviewed journal and conference papers published over the last five years (or since last promotion or initial appointment, whichever is less) related to pedagogy or other relevant areas of education. Include the journal's impact factor (or equivalent journal ranking data) and the number of citations of the article (s)" (*Ohio University, 2015*).

### Not all mentions of the JIF support its use

While the majority of the mentions found in our sample of RPT documents were either neutral or supportive of the JIF, we find that 13% of institutions had at least one mention which cautioned against or discouraged use of the JIF in evaluations. We observed varying levels of caution in these mentions. Some do not critique use of the JIF in general, but rather express concern that JIF data are not as relevant for their discipline as for others. For example, criteria for promotion and tenure from the School of Social Work at the University of Central Florida state: "Journal impact factors will not be a primary criteria for the measurement of scholarly activity and prominence as the academic depth and breadth of the profession requires publication in a multitude of journals that may not have high impact factors, especially when compared to the stem [sic] disciplines" (*University of Central Florida, 2014*).

Similarly, guidelines from the Department of Human Health and Nutritional Sciences at the University of Guelph call the JIF a "problematic" index and discourage its use while again highlighting disciplinary differences: "Discussion of journal quality (by those familiar with the field) may be included in the assessment in addition to consideration of the quality of individual research contributions. However, citation analyses and impact factors are problematic indices, particularly in comparisons across fields, and their use in the review process is not encouraged" (*University of Guelph, 2008*).

Other guidelines, such as those from the Faculty of Veterinary Medicine at the University of Calgary, caution against relying solely on the JIF as a measure of quality, but still allow it to be considered: "Special consideration is to be given to the quality of the publication and the nature of the authorship. Contributions of the applicant must be clearly documented. The reputation and impact of the journal or other publication format will be considered, but takes secondary consideration to the quality of the publication and the nature of the contributions. Impact factors of journals should not be used as the sole or deciding criteria in assessing quality" (*University of Calgary, 2008*).

Some RPT documents even seem to show disagreement within evaluation committees on the use of the JIF. For example, a document from the Committee on Academic Personnel at the University of California, San Diego reads: "CAP [Committee on Academic Personnel] welcomes data on journal acceptance rates and impact factors, citation rates and H-index, but some CAP members (as do senior staff of scholarly societies) retain various degrees of skepticism about such measures" (*University of California, San Diego, 2016*).

None of the RPT documents we analyzed heavily criticize the JIF or prohibit its use in evaluations.

## Discussion

To our knowledge, this is the first study of RPT documents from a representative sample of US and Canadian universities to analyze the use of the JIF in academic evaluations. We found that 40% of R-type and 18% of M-type institutions mentioned the JIF or related terms in their RPT documents. Mentions were largely supportive of JIF use, with 87% of institutions having at least one supportive mention, while just 13% had cautious mentions. The most common specified association we observed in these documents was between the JIF and quality.

## How prevalent is the use of the JIF in evaluations?

Mentions of the JIF and related terms in RPT documents are not as ubiquitous as the amount of discussion of current evaluation systems would suggest – 23% of institutions in our sample used these terms explicitly. Sample considerations, including the relatively small total number of institutions included, could be a factor in calculating the prevalence of the use of the JIF. However, given our stratified random sampling approach, we consider our sample to be representative and a good indicator of what would be found in the larger population of U.S. and Canadian universities. Importantly, we note that the results differ depending on institution type, which might suggest that the experiences at R-type universities (where mentions of the JIF were most prevalent) play an outsized role in discussions about evaluation. Furthermore, the analysis we present on the terms in groups 1 and 2 of our coding terminology (see *Figure 1*) may represent only the tip of the iceberg. That is, while we analyzed only those terms that were very closely related to the JIF, we also observed (but did not analyze) terms such as 'major', 'prestigious', 'prominent', 'highly respected', 'highly ranked', and 'top tier' that may be associated with high JIFs in the minds of evaluators. It is impossible to know how RPT committee members interpret such phrases on the basis of the documents alone, but we suspect that some of these additional terms serve to invoke the JIF without explictly naming it. Take the following example from the Department of Anthropology at Boise State University that leaves open for interpretation what measure is used for determining a journal's status (emphasis added): "The candidate for promotion to associate rank should have a least two publications in **upper-tier journals**".

Such examples do not explicitly mention the JIF (and thus are not counted in our analysis), but do imply the need for some measure for ranking journals. It seems likely, given the ubiquity of the JIF, that some committee members will rely on this metric, at least in part, for such a ranking. In short, counting mentions of a restricted set of terms, as we have done here, is likely an underestimate of the extent of the use of the JIF in RPT processes. However, we believe the in-depth analysis presented herein provides a glimpse into the current use of the JIF and may indicate how faculty are considering the metric in evaluations, particularly with respect to assessments of quality.

## The JIF does not measure quality

The association between the JIF and quality was found in 63% of institutions in our sample, but is there evidence that the JIF is a good indicator of quality? Although quality is hard to define, and even harder to measure, there are aspects of methodological rigor which could be considered indicative of quality, such as sample size, experimental design, and reproducibility (*Brembs, 2018*). What is the relationship between these aspects of a study and the JIF?

Evidence suggests that methodological indicators of quality are not always found in journals with high JIFs. For example, *Fraley and Vazire (2014)* found that social and personality psychology journals with the highest JIFs tend to publish studies with smaller sample sizes and lower statistical power. Similarly, *Munafò et al. (2009)* report that higher-ranked journals tend to publish gene-association studies with lower sample sizes and overestimate effect sizes. Analyses of neuroscience and/or psychology studies show either no correlation (*Brembs et al., 2013*) or a negative correlation (*Szucs and Ioannidis, 2017*) between statistical power and the JIF.

Several studies have looked at experimental design to assess methodological rigor and quality of a study. *Chess and Gagnier (2013)* analyzed clinical trial studies for 10 different indicators of quality, including randomization and blinding, and found that less than 1% of studies met all 10 quality criteria, while the JIF of the journals did not significantly predict whether a larger number of quality criteria were met. *Barbui et al. (2006)* used three different scales that take into account experimental design, bias, randomization, and more to assess quality, and found no clear relationship between the JIF and study quality.

Reproducibility could be used as a measure of quality, since it requires sufficient methodological care and detail. *Bustin et al. (2013)* analyzed molecular biology studies and found key methodological details lacking, reporting a negative correlation between the JIF and the amount of information provided in the work. *Mobley et al. (2013)* found that around half of biomedical researchers surveyed reported they were unable to reproduce a published finding, some from journals with a JIF over 20. *Prinz et al. (2011)* found "that the reproducibility of published data did not significantly correlate with journal impact factors" (pg. 2).

Thus, at least as viewed through the aspects above, there is little to no evidence that the JIF measures research quality. For a more comprehensive review, see *Brembs (2018)*.

### Improving academic evaluation

In the last few years, several proposals and initiatives have challenged the use of the JIF and promoted the responsible use of metrics to improve academic evaluations. These include the Leiden Manifesto (*Hicks et al., 2015*), the Metric Tide report (*Wilsdon et al., 2015*), the Next-Generation Metrics report (*Wildson et al., 2017*), and HuMetricsHSS (humetricshss.org). We provide a brief description of some such efforts (for a review, see *Moher et al., 2018*).

Probably the most well-known initiative is the Declaration on Research Assessment (DORA; sfdora.org). DORA outlines limitations of the JIF, and puts forward a general recommendation to not use the JIF in evaluations, especially as a "surrogate measure of the quality of individual research articles" (sfdora.org/read). Particularly relevant to our current research is DORA's recommendation for institutions to "be explicit about the criteria used to reach hiring, tenure, and promotion decisions, clearly highlighting. . .that the scientific content of a paper is much more important than publication metrics or the identity of the journal in which it was published." DORA's new strategic plan (*DORA Steering Committee, 2018*) includes spreading awareness of alternatives to the JIF and collecting examples of good evaluation practices (sfdora.org/good-practices). To date, DORA has been signed by over 1400 organizations and 14,000 individuals worldwide. None of the institutions in our sample are DORA signatories, but it would be interesting to study how commitment to DORA might be reflected in changes to an institution's RPT documents and evaluations.

Libraries are leaders in promoting the responsible use of metrics, developing online guides (see, for example, *Duke University Medical Center Library & Archives, 2018*; *University of Illinois at Urbana Champaign Library, 2018*; *University of Surrey Library, 2018*; *University of York Library, 2018*), and providing in-person advising and training for faculty in publishing and bibliometrics. The Association of College and Research Libraries (ACRL) has developed a Scholarly Communication Toolkit on evaluating journals (*Association of College & Research Libraries, 2018*), which outlines ways to assess journal quality that go beyond

metrics like the JIF. LIBER (Ligue des Bibliothèques Européennes de Recherche) has established a working group which recently recommended increased training in metrics and their responsible uses (*Coombs and Peters, 2017*). The Measuring your Research Impact (MyRI) project (myri.conul.ie) is a joint effort by three Irish academic libraries to provide open educational resources on bibliometrics. The Metrics Toolkit (www.metrics-toolkit.org) is a collaborative project by librarians and information professionals to provide "evidence-based information" on traditional and alternative metrics, including use cases.

## Conclusions

Overall, our results support the claims of faculty that the JIF features in evaluations of their research, though perhaps less prominently than previously thought, at least with respect to formal RPT guidelines. Importantly, our analysis does not estimate use of the JIF beyond what is found in formal RPT documents, such as faculty members who serve on review committees and pay attention to this metric despite it not being explicitly mentioned in guidelines. Future work will include surveying faculty members, particularly those who have served on RPT committees, to learn more about how they interpret and apply RPT guidelines in evaluations and investigate some of the more subjective issues not addressed in this study.

Our results also raise specific concerns that the JIF is being used to evaluate the quality and significance of research, despite the numerous warnings against such use (*Brembs et al., 2013*; *Brembs, 2018*; *Moustafa, 2015*; *Haustein and Larivière, 2015*; *Sugimoto and Larivière, 2018*; *Seglen, 1997*; *Kurmis, 2003*; *The Analogue University, 2019*). We hope our work will draw attention to this issue, and that increased educational and outreach efforts, like DORA and the library-led initiatives mentioned above, will help academics make better decisions regarding the use of metrics like the JIF.

### Acknowledgements

We are grateful to SPARC, the OpenCon community, the DORA Steering Committee (especially Catriona MacCallum and Anna Hatch), Chealsye Bowley, and Abigail Goben for discussions that shaped and improved this work. We also thank Elizabeth Gadd and Erika Mias, who suggested library guides and projects on

responsible metrics to highlight in our Discussion.

**Erin C McKiernan** is in the Departamento de Física, Universidad Nacional Autónoma de México, Mexico City, Mexico

emckiernan@ciencias.unam.mx

https://orcid.org/0000-0002-9430-5221

**Lesley A Schimanski** is in the Scholarly Communications Lab, Simon Fraser University, Vancouver, Canada

https://orcid.org/0000-0002-4664-179X

**Carol Muñoz Nieves** is in the Scholarly Communications Lab, Simon Fraser University, Vancouver, Canada

https://orcid.org/0000-0002-8857-3000

**Lisa Matthias** is at the John F Kennedy Institute, Freie Universität Berlin, Berlin, Germany

https://orcid.org/0000-0002-2612-2132

**Meredith T Niles** is in the Department of Nutrition and Food Sciences, University of Vermont, Burlington, United States

https://orcid.org/0000-0002-8323-1351

**Juan P Alperin** is in the School of Publishing and the Scholarly Communications Lab, Simon Fraser University, Vancouver, Canada

juan@alperin.ca

https://orcid.org/0000-0002-9344-7439

*Author contributions:* Erin C McKiernan, Conceptualization, Data curation, Formal analysis, Funding acquisition, Methodology, Writing—original draft, Writing—review and editing; Lesley A Schimanski, Conceptualization, Formal analysis, Methodology, Project administration, Writing—review and editing; Carol Muñoz Nieves, Conceptualization, Data curation, Formal analysis, Methodology, Writing—review and editing; Lisa Matthias, Formal analysis, Writing—review and editing; Meredith T Niles, Conceptualization, Funding acquisition, Writing—review and editing; Juan P Alperin, Conceptualization, Resources, Data curation, Software, Formal analysis, Supervision, Funding acquisition, Methodology, Writing—original draft, Project administration, Writing—review and editing

*Competing interests:* Erin C McKiernan: is a member of the DORA Steering Committee and an advisor for the Metrics Toolkit, both volunteer positions. The other authors declare that no competing interests exist.

## Funding

| Funder | Grant reference number | Author |
|---|---|---|
| Open Society Foundations | OR2016-29841 | Erin C McKiernan Meredith T Niles Juan Pablo Alperin |

The funders had no role in study design, data collection and interpretation, or the decision to submit the work for publication.

### Decision letter and Author response

Decision letter https://doi.org/10.7554/eLife.47338.009
Author response https://doi.org/10.7554/eLife.47338.010

## Additional files

### Supplementary files

• Supplementary file 1. Supplemental information.
DOI: https://doi.org/10.7554/eLife.47338.003

• Transparent reporting form
DOI: https://doi.org/10.7554/eLife.47338.004

### Data availability

The data that support the findings of this study are available in the Harvard Dataverse with the identifier https://doi.org/10.7910/DVN/VY4TJE (Alperin et al., 2018). These data include the list of institutions and academic units for which we have acquired documents along with an indicator of whether terms related to the impact factor were found in the documents for the institution or academic unit, as well as the qualitative coding of each mention reported.

The following previously published dataset was used:

| Author(s) | Year | Dataset URL | Database and Identifier |
|---|---|---|---|
| Alperin JP, Muñoz Nieves C, Schimanski L, McKiernan EC, Niles MT | 2018 | https://doi.org/10.7910/DVN/VY4TJE | Harvard Dataverse, 10.7910/DVN/VY4TJE |

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
