## [Decision Letter]

Thank you for submitting your article "Use of the Journal Impact Factor in academic review, promotion, and tenure evaluations" for consideration by *eLife*. Your article has been reviewed by two peer reviewers, and the evaluation has been overseen by myself (Emma Pewsey, Associate Features Editor) and Peter Rodgers (Features Editor). The following individual involved in review of your submission has agreed to reveal his identity: Björn Brembs (Reviewer #1).

The reviewers have discussed the reviews with one another and I have drafted this decision to help you prepare a revised submission.

Summary:

By ascertaining how often review, promotion and tenure committees refer to the journal impact factor (JIF), this paper provides solid data about an issue that is often discussed using only subjective experience, hearsay and estimates. The paper is well written and easy to read, and is an important contribution to the discourse on organizing the way we go about scholarship.

Essential revisions:

1) In the Materials and methods section, please provide the percentage of respondents, and provide the sample size as a percentage of number of US/Canada institutions total and sampled.

2) An explanation should be added to the Materials and methods section to explain why, with more than 5,000 universities and colleges in the US alone, the sample of 129 universities counts as a "representative" sample, and explain how the numbers were achieved.

3) Please also discuss the size of the sample in the Discussion section.

4) The importance of this study would be significantly greater if JIF usage could be associated with RTP promotion outcome. Without that information the importance of this study is limited to documenting with data what most people already think that they know.

---

## [Author Response]

Essential revisions:1) In the Materials and methods section, please provide the percentage of respondents, and provide the sample size as a percentage of number of US/Canada institutions total and sampled.

To address this concern, we reproduced a simplified version of Table 1 from our previous paper, which gives an overview of the population of universities from the United States and Canada by type, the number and percent randomly chosen for our stratified sample, and the number of institutions for which we obtained at least one relevant document. We added a reference to this table at the end of the first paragraph of the subsection “Document collection”, along with a sentence describing sample considerations (see point 2 below).

2) An explanation should be added to the Materials and methods section to explain why, with more than 5,000 universities and colleges in the US alone, the sample of 129 universities counts as a "representative" sample, and explain how the numbers were achieved.

As per the table added, we also included a sentence that describes why we consider this sample to be representative. The sentence, added in the first paragraph of the “Document Collection” subsection, reads as follows: “The stratified sample was designed to collect documents from enough institutions in each of the R-, M-, and B-type categories to have a statistical power of. 8 (assuming a small effect size of. 25 of a standard deviation) when making comparisons between disciplines.”

3) Please also discuss the size of the sample in the Discussion section.

We have added a few sentences about sample size considerations to the Discussion section on the prevalence of the use of the JIF in evaluations.

4) The importance of this study would be significantly greater if JIF usage could be associated with RTP promotion outcome. Without that information the importance of this study is limited to documenting with data what most people already think that they know.

We appreciate the reviewers’ concerns here. Unfortunately, we have no way of obtaining information on the outcomes of RTP evaluations, especially because this information is often privileged and not publicly documented by departmental committees. However, we believe that our study is still important, even if some of the results do document what people already think they know about RPT evaluations. The information previously available was largely based on small surveys or largely anecdotal. Our study is the first to provide concrete data on how often and in what ways the JIF is used in formal documents governing RPT processes. This information could be valuable to researchers, administrators, and others seeking to understand current issues with RPT processes and how to improve them.